# A Case Study of Chimeric Antigen Receptor T Cell Function: Donor Therapeutic Differences in Activity and Modulation with Verteporfin

**DOI:** 10.3390/cancers15041085

**Published:** 2023-02-08

**Authors:** Jiyong Liang, Dexing Fang, Joy Gumin, Hinda Najem, Moloud Sooreshjani, Renduo Song, Aria Sabbagh, Ling-Yuan Kong, Joseph Duffy, Irina V. Balyasnikova, Seth M. Pollack, Vinay K. Puduvalli, Amy B. Heimberger

**Affiliations:** 1Department of Neurosurgery, The University of Texas MD Anderson Cancer Center, Houston, TX 77030, USA; 2Department of Neuro-Oncology, The University of Texas MD Anderson Cancer Center, Houston, TX 77030, USA; 3Department of Neurological Surgery, Feinberg School of Medicine, Northwestern University, Chicago, IL 60611, USA; 4Malnati Brain Tumor Institute of the Lurie Comprehensive Cancer Center, Feinberg School of Medicine, Northwestern University, Chicago, IL 60611, USA; 5Department of Cancer Biology, Feinberg School of Medicine, Northwestern University, Chicago, IL 60611, USA; 6Department of Neurosurgery, Northwestern University, Simpson Querrey Biomedical Research Center, 303 E. Superior Street, 6-516, Chicago, IL 60611, USA

**Keywords:** trogocytosis, epidermal growth factor receptor variant III, chimeric antigen receptor T cells, autophagy, phagocytosis, glioblastoma

## Abstract

**Simple Summary:**

The loss of tumor antigens prevents the immune system from recognizing and destroying cancer cells. Immune cells can remove these antigens and express them on their surface. Other immune cells becoming confused, kill the anti-tumor immune cells. By blocking this process using a drug commonly used to treat a variety of eye conditions, we were able to restore anti-tumor immune responses for impaired T cells in mouse models of brain cancer.

**Abstract:**

Background: Chimeric antigen receptor (CAR) T cells have recently been demonstrated to extract and express cognate tumor antigens through trogocytosis. This process may contribute to tumor antigen escape, T cell exhaustion, and fratricide, which plays a central role in CAR dysfunction. We sought to evaluate the importance of this effect in epidermal growth factor receptor variant III (EGFRvIII) specific CAR T cells targeting glioma. Methods: EGFRvIII-specific CAR T cells were generated from various donors and analyzed for cytotoxicity, trogocytosis, and in vivo therapeutic activity against intracranial glioma. Tumor autophagy resulting from CAR T cell activity was evaluated in combination with an autophagy inducer (verteporfin) or inhibitor (bafilomycin A1). Results: CAR T cell products derived from different donors induced markedly divergent levels of trogocytosis of tumor antigen as well as PD-L1 upon engaging target tumor cells correlating with variability in efficacy in mice. Pharmacological facilitation of CAR induced-autophagy with verteporfin inhibits trogocytic expression of tumor antigen on CARs and increases CAR persistence and efficacy in mice. Conclusion: These data propose CAR-induced autophagy as a mechanism counteracting CAR-induced trogocytosis and provide a new strategy to innovate high-performance CARs through pharmacological facilitation of T cell-induced tumor death.

## 1. Introduction

Trogocytosis is a process whereby cell surface molecules are transferred to effector T cells through the immunologic synapse, often by antigen-presenting cells (APCs). While this process has long been recognized as a critical aspect of functional immunity [1], it has recently emerged as a potentially important cause of chimeric antigen receptor (CAR) T cell dysfunction [2,3]. CAR T cells are generated by transducing a T cell with a CAR construct containing a tumor antigen recognition domain linked to the constant regions of a signaling T cell receptor. The CAR T cell then recognizes the tumor antigen with high specificity in a non-MHC-restricted manner that is independent of antigen processing [4]. During CAR-induced trogocytosis, the CAR T cells can uptake cognate antigens from the surface of target cells, allowing the escape of target cells through localized antigen loss at the immunologic synapse and fratricide of CAR T cells once target antigen becomes expressed on the CAR T cell surface [2]. Overstimulation of the effector CAR T cells may also lead to the development of an exhausted phenotype.

We have been developing EGFRvIII-specific CAR T cells for the treatment of glioma and hypothesize that trogocytosis could potentially play a role in the development of antigen loss, and T cell exhaustion previously reported [5,6]. CAR T cell-mediated tumor killing is often oversimplified as instantaneous perforin-induced lysis of tumor cells. In fact, there is a complex process of autophagy induced by T cells [7], which may actually oppose trogocytosis by mediating the degradation of endocytic substrates in tumor targets. Previously, we demonstrated that verteporfin acts as an autophagy inducer and promotes autophagosome formation and autophagy flux, as shown by increases in LC3-II, decreases in p62, and autophagy-mediated degradation [8]. The purpose of this study was to ascertain if pharmacological facilitation of autophagy with verteporfin inhibits trogocytosis of the EGFRvIII tumor antigen by CAR T cells and to determine if this increases their persistence and efficacy in mice. 

## 2. Materials and Methods

### 2.1. Cell Lines

The U87 cell line was obtained from the American Type Culture Collection. The K562 EGFRvIII clone 27 (activating and antigen-presenting cells, aAPCs) with stable expression of 41BB-L, CD86, CD64, tCD19, and membrane-tethered IL-15 was a gift from Dr. Laurence Cooper at MD Anderson Cancer Center. K562 cells were maintained in RPMI1640 supplemented with 10% fetal bovine serum (FBS). U87-EGFRvIII-Zeomycin cells were a gift from Dr. Oliver Bogler at MD Anderson Cancer Center and were cultured in complete Dulbecco’s Modified Eagle Medium (DMEM) containing 10% FBS and 2mM glutamax at 37 °C and in an atmosphere of 95% air/5% CO_2_.

### 2.2. EGFRvIII CAR Engineering

As previously described [5], the EGFRvIII CAR was engineered by fusing the CSF2RA signal peptide, scFvs of monoclonal antibodies 139 with a Whitlow linker inserted between the light and heavy chains, the IgG4 extracellular stalk, and the CD28 and CD3-zeta intracellular signaling domains. The plasmid encoding for CAR was then co-transfected with the Sleeping Beauty (SB) plasmid. CAR+ T cells were stimulated in a 1:2 ratio with the irradiated EGFRvIII+ K562 in the presence of 30 ng/mL IL-21. For in vivo trafficking, Firefly luciferase was cloned in frame to the C terminus of the EGFRvIII CAR construct with a P2A self-cleavage linker. Human donor T cells were transfected by electroporation with the CAR-P2A-ffLUC SB transposon along with the SB11 transposase.

### 2.3. T Cells Isolation, Transfection, and Ex Vivo CAR Expansion

Peripheral blood mononuclear cells (PBMCs) from healthy donors were obtained from the Gulf Coast Regional Blood Bank and isolated using Ficoll-Paque (GE Healthcare, Chicago, IL, USA) per the manufacturer’s protocol. The PBMCs were either freshly used or cryopreserved and thawed immediately before use. CD3^+^ T cell selection was performed using the Human Pan T cell isolation microbeads (Miltenyi Biotec, San Diego, CA) per the manufacturer’s protocol. The cells were then allowed to rest for 2 h before CAR nucleofection using Amaxa Nucleofector 2B (Lonza, Basel, Switzerland). Briefly, 20 × 10^6^ CD3^+^ cells were suspended in 100 µL of human T cell electroporation buffer (Lonza, Basel, Switzerland), 10 µg of CAR plasmid, and 5 µg of SB11 transposase (a gift from Amer Najjar at the MD Anderson Cancer Center) and then electroporated using program U-014. Immediately after electroporation, the cells were transferred to a prewarmed recovery medium (phenol-free RPMI medium, 20% FBS, and 2 mM glutamax) for 24 h. The cells were then counted and phenotyped by flow cytometry to determine the percentage of CAR expression. Cells were harvested for initial stimulation with 100 Gy-irradiated aAPCs at a 1:2 ratio (CAR^+^: aAPC) in the presence of 30 ng/mL IL-21 in RPMI1640 medium supplemented with 10% FBS. Cytokines were replenished every other day for 7 days. Subsequent expansion after 7 days was performed with a weekly iteration of aAPC stimulation and the addition of 50 U/mL IL-2 and 30 ng/mL IL-21 every other day.

### 2.4. Glioma and CAR T Cell Co-Culture

Ex vivo expanded EGFRvIII CAR T cells were added to target (U87-EGFRvIII) glioma cells at indicated cell ratios.

### 2.5. Flow Cytometry Analysis and Trogocytosis Assay

For direct flow cytometry, up to 10^6^ cells were stained with mAbs (Appendix A) in FACS buffer (PBS, 2% FBS, 0.05% sodium azide) for 30 min in the dark at 4 °C. Data were collected on a FACS Celesta (BD Biosciences, San Jose, CA, USA) using the FACSDiva software (version 8.0.1, BD Biosciences, San Jose, CA, USA). All data were analyzed using FlowJo software (version 10.7, TreeStar, Ashland, OR, USA). CAR T cells were exposed to target tumor cells at a 1:1 ratio and then harvested and probed with an EGFRvIII primary antibody (V3980, NSJ Bioreagents, San Diego, CA, USA) and a secondary antibody tagged with PE or APC followed by flow cytometry analysis.

### 2.6. Evaluation of T Cell Efficacy in Intracranial Glioma Xenografts

Animal experiments were carried out according to regulations from the Institutional Animal Care and Use Committee (IACUC) at MD Anderson Cancer Center (ACUF 00001544-RN00). Both female and male NOD.Cg-PrkdcscidIL2Rγtm1Wjl/Sz (NSG, Jackson Laboratory, Bar Harbor, ME) mice aged 6–8 weeks were anesthetized by intraperitoneal injection using a cocktail of 10 mg/mL ketamine and 0.5 mg/mL xylazine at a dose of 0.1 mL/10 g. A guide screw was surgically implanted 2.5 mm to the right of the coronal suture and 1 mm posterior to the bregma at a depth of 3 mm. Two weeks after surgery, intracranial tumors were established by implantation of 250,000 U87-EGFRvIII-Zeomycin in 5 μL PBS through the guide screw. The CAR infiltration in the tumor was serially imaged using Xenogen IVIS Spectrum (Caliper Life Sciences, Hopkinton, MA, USA) 10 min after intraperitoneal injection of 3 mg D-luciferin potassium salt (cat. # MB000102-R70170, Syd Labs, Natick, MA). CAR flux (photons/s/cm^2^/steradian) was measured using Living Image software (version 2.50, Caliper Life Sciences, Hopkinton, MA, USA) in a delineated region encompassing the entire cranium. Mice were sorted into treatment groups 4 days after tumor implantation. The mice were treated on day 5 with 4 × 10^6^ EGFRvIII CAR T cells in 5 μL of PBS administered intracranially through the guide screw and treated per the designated schemas. Mice were treated with verteporfin (10 mg/Kg) via intraperitoneal injection. Mice were sacrificed when they displayed progressive weight loss of >25%, rapid weight loss of >10% within 48 h, hind limb paralysis, or any two of the following clinical symptoms of illness: ataxia, hunched posture, or irregular respiration rate.

### 2.7. Statistical Analyses

Statistical analyses were performed in GraphPad Prism, version 8.0. Data were examined for normality through boxplots and QQ plots prior to the application of linear statistical models. Statistical analyses of in vitro assays were performed using one-way ANOVA, two-way ANOVA, or unpaired T-test, as indicated in the figure legends. Analyses of in vivo tumor BLI imaging were performed using two-way ANOVA with repeated measures and Sidak’s post hoc test for multiple comparisons. Survival analysis of the mice was performed using the log-rank (Mantel-Cox) test. Significance of the findings is defined as follows: ns = not significant, *p* >= 0.05; * *p* < 0.05; ** *p* < 0.01; *** *p* < 0.001; **** *p* < 0.0001. 

## 3. Results

### 3.1. Differential Trogocytosis and Efficacy in CAR Products

Our group has a long-standing interest in trying to analyze why different donors exhibit different CAR T cell activity despite identical cell manufacturing procedures using the same CAR construct. We hypothesized that trogocytosis might play a differential role in CAR T cell dysfunction and contribute to donor differences in CAR activity. As part of this effort, we sought to identify two dichotomized CAR T cell products that have significant differences in trogocytosis. Much higher levels of EGFRvIII were detected on the surface of CAR T cells of Donor 1 relative to other donors (Figure 1A). As a control, we confirmed that EGFRvIII was detected abundantly at basal levels on target tumor cells expressing EGFRvIII but not on antigen naïve CARs (Figure 1B).

To further characterize two dichotomized CAR T cell products, two donors were selected and labeled “donor 1” and “donor 2.” Donor 1 CAR T cells resulted in minimal therapeutic effect in vivo compared with the relatively higher level of potency of Donor 2 (40% long-term survivors) (Figure 2A), as well as in the U87EGFRvIII tumor line in vitro (Figure 2B; *p* = 0.0001).

To ascertain if the phenotypic composition might account for the difference between products, the CAR product was profiled. Both donors exhibited CAR lineage skewing towards the CD8+ population with an expansion-dependent loss of the CD4+ population. This skewing was observed regardless of initial CD8:CD4 ratios (Figure 3A), suggesting that T cell phenotype was not a key variable explaining their differences in therapeutic response. Furthermore, CARs of both donors expressed T cell exhaustion markers PD-1, TIM-3, and LAG-3 on day 28 of ex vivo expansion, with Donor 2 CAR T cells expressing relatively higher levels of these markers despite their higher therapeutic efficacy (Figure 3B,C). In contrast, Donor 1, but not Donor 2 CAR T cells, expressed CD57, the T cell senescence marker, at a higher level (Figure 3B,C). Thus, expansion-induced CAR lineage skewing and expression of exhaustion markers alone were insufficient to explain donor variations in CAR efficacy.

### 3.2. Autophagy Antagonizes Trogocytosis and Increases Target Killing

The trogosome cargos are likely processed through the endosome systems, where cargo is either recycled to the cell surface or sorted for lysosomal and autophagy-dependent degradation (Figure 4). Through assessment of the trogocytosis kinetics, we found expression of EGFRvIII on CAR T cells was maximal at 2 h and declined by 4 h (Figure 5A). We recently reported that verteporfin activates the autophagy pathway and that PD-L1 is degraded as part of this process [8]. We hypothesized that verteporfin might induce autophagy to counter trogocytosis through increased degradation of EGFRvIII. Indeed, we found the level of EGFRvIII expression was reduced in co-cultures treated with verteporfin (Figure 5B).

Next, we assayed the effect of verteporfin treatment on CAR-mediated target cell killing. Verteporfin, whether alone or in co-treatment with CARs, was found to have no effect on target cell viability at lower E:T ratios. However, at a ratio of 5:1 E:T, Donor 1 CAR killing ability was markedly increased when treated with verteporfin (Figure 5C). Verteporfin did not affect the cytotoxic activity of Donor 2 CARs (Figure 5D) that had lower levels of trogocytosis.

### 3.3. CAR-Induced Autophagy Mediates Degradation of Cross-Transferred PD-L1

Because the immune checkpoint ligand PD-L1 and other proteins can be transferred to T cells and monocytes from antigen-presenting and cancer cells [9,10], we next evaluated whether such proteins were transferred to the CAR T cells in our system. At baseline, PD-L1 is not expressed on primary T cells (Figure 6A), on CAR T cells expanded on PD-L1-feeders (Figure 6B), or on anti-CD3/CD28 dynabeads (Figure 6C). Although PD-L1 is almost never observed on circulating T cells in any normal physiologic circumstance, there are some circumstances where this has been observed [11,12]. To be sure that the PD-L1 expression on the CAR T effectors was indeed a product of trogocytosis rather than a result of an inducible mechanism (such as IFN-γ) that can sometimes be responsible for increases in PD-L1 expression in target tumor cells, expression profiling was performed in the setting of IFN-γ. As would be expected, PD-L1 was not detected on CAR T cells either treated with IFN-γ or following exposure to CAR-target condition media in the absence of targets (Figure 6C) and was only detected on CARs in the presence of targets (Figure 6D). To examine the impact on trogocytic PD-L1 uptake by CAR T cells, we inhibited autophagy using the drug bafilomycin. Indeed, enhanced PD-L1 uptake was observed with its use (Figure 6E). Furthermore, bafilomycin-related autophagy inhibition was also more potent in Donor 2 compared to Donor 1 (Figure 6E), suggesting counteracting mechanisms of trogocytosis and autophagy (Figure 4).

### 3.4. Verteporfin Increases CAR T Cell Persistence and Efficacy

To improve CAR persistence in the GBM tumor microenvironment, we sought to determine whether autophagy counters trogocytosis in vivo. We first assayed the abundance of CAR T cells in the brain of U87-EGFRvIII tumor-bearing mice treated with or without verteporfin. The bioluminescence (BLI) CAR T cell signal was only detected in CAR-infused tumors (Figure 7A). Although there was no significant difference in CAR retention noted one day after infusion, verteporfin-treated mice maintained significantly higher levels of CAR signal ten days after infusion (Figure 7B; *p* < 0.0001). 

We hypothesized that the impact of verteporfin on improving CAR T cell function and persistence is independent of any activity it might exert on the tumor cells themselves in vivo. To test this, we evaluated the dysfunctional Donor 1 CAR T cells in combination with verteporfin. Consistent with our previous study [8], verteporfin alone did not exert any therapeutic effect (Figure 8A). However, treatment with the combination of verteporfin and Donor 1 CAR T cells extended the survival of EGFRvIII^+^ tumor-bearing mice relative to both PBS and monotherapy with CAR T cells (Figure 8B; *p* = 0.02).

## 4. Discussion

Trogocytosis is a newly recognized mechanism underlying CAR T cell dysfunction leading to tumor antigen escape, T cell exhaustion, and CAR T cell fratricide [2]. Here we confirm that trogocytosis occurs in the stimulation of EGFRvIII CAR T cells but surprisingly find that it is variable between donors. As might be expected, we find that the amount of trogocytosis is inversely associated with CAR efficacy in a murine glioma model. Notably, CAR-induced trogocytosis also mediates the cross-transfer of the immune checkpoint ligand PD-L1 in addition to tumor antigens in this system. Thus, efforts to reduce trogocytosis may enhance CAR function and improve efficacy. Through activation of CAR T cell-induced autophagy, verteporfin inhibits trogocytosis, increases CAR persistence in vivo, and improves the efficacy of CAR T cells targeting EGFRvIII+ tumors in vivo. CAR expression levels have been previously found to be inversely associated with CAR function, with higher CAR expression increasing CAR signaling and dysfunction [13,14,15,16]. Consistent with these findings, CAR levels were found to be lower in representative Donor 1 relative to Donor 2, which may contribute to increased CAR activity and be linked to differences in trogocytosis, which is a CAR-dependent process. Furthermore, this study suggests that varying amounts of trogocytosis following CAR T cell target recognition is an important factor creating differences in CAR T cell function and therapeutic efficacy between donors.

Other pharmacological strategies that could be considered for the inhibition of trogocytosis include phosphoinositide 3-kinase (PI3K) inhibitors [2,17,18]. However, non-selective effects on T cell viability and function may be problematic. More selective PI3K inhibitors may resolve this problem [19,20]. Ultimately genetic modulation of this pathway in CAR T cells will likely be the path forward. Our findings suggest that trogocytosis can be counteracted by verteporfin-facilitated autophagy, pushing the downstream pathway into late-endosome formation and degradation of the target cell antigens instead of its re-expression on the cell surface by recycling endosomes (Figure 4), offering a strategy to circumvent some of the limitations associated with targeting trogocytosis. Although our findings might be specific to the model system used in this study, both trogocytosis and autophagy are universal cell processes. Thus, the specific use of verteporfin for other CAR T cells, such as those targeting IL13RA2, GD2 [21], and others [22,23], will need to be tested to validate these findings. 

T cells were isolated and expanded from healthy donors and then engineered to express the full CAR construct. This CAR system requires the presence of a stimulating cell line for CAR T cell proliferation and expansion [24]. The CAR expression in Donor 1 was higher than in Donor 2. However, in vivo cytotoxicity of the CAR T cells from Donor 2 was higher than Donor 1. It should be noted that higher levels of CAR expression are not always associated with increased CAR function. Instead, higher CAR expression can be associated with tonic CAR signaling, which causes exhaustion and impairs CAR T cell function and persistence [13,14,15,16]. Our findings appear to be consistent with these data.

Although tumor antigens seem particularly susceptible to transfer through CAR-induced trogocytosis [2], the full extent of adjacent membrane proteins transferred through the trogosome remains undetermined [25,26,27,28] because, in the CAR system we used, a non-targeting CAR T cell as a control was not feasible. As such, we evaluated whether other targets could be involved in trogocytosis. In addition to CAR-induced trogocytosis of cognate tumor antigen, we documented cross-cell transfer of the immune checkpoint ligand PD-L1 from tumor cells to CAR T cells. This observation may be a function of verteporfin regulating the turnover of receptors more generally. The acquisition of PD-L1 in T cells has previously been shown to inhibit effector function in circulating T cells [11,12]. To our knowledge, this is the first description of PD-L1 expression on CAR T cells acquired through trogocytosis. The functional immunological consequences of this finding will be a focus of future studies. Since PD-1 is overexpressed by exhausted T cells, its engagement with PD-L1 expressing CAR T cells is expected to activate the immune checkpoint and may also represent an underlying mechanism of fratricide as a consequence of CAR-induced trogocytosis. 

While trogocytosis is markedly different between these two donors, there are undoubtedly many factors that may explain the difference in the effector functions of CAR-T cells identified between these two donors, including promotor polymorphism of immune stimulatory cytokines and HLA matching. As opposed to standard CAR-T cell therapies that are used clinically, this model is an allogeneic therapy in an immunocompromised mouse model. The endogenous T cell receptor for donor 1 and donor 2 have not been deleted, so differential alloreactivity between these two donors is a confounder for analysis. While this first study establishes the phenomenon, a larger study with more donors will be needed to estimate exactly how big a factor trogocytosis is in CAR T product variability. It is also important to note that verteporfin has multiple biological and cellular effects. It is also possible that verteporfin is altering CAR turnover or proliferation. As such, it is unclear if the results are exclusively due to verteporfin via its effect on trogocytosis. Additional mechanistic studies are required to elucidate exactly how autophagy activation by verteporfin is associated with the inhibition of trogocytosis. Moreover, there are intrinsic effects of verteporfin on glioma cells [29]. Ultimately, strategies that genetically manipulate these functions in CAR-T cells will likely be evaluated and may be the preferred approach moving forward. Notably, there are additional hurdles that will need to be considered that influence CAR anti-tumor activity, such as distribution in the tumor microenvironment and tumor-mediated immune suppression [30,31]. 

Our prior study showed that the photodynamic agent verteporfin induces autophagy and selective degradation of the PD-L1 immune checkpoint ligand at clinically achievable concentrations [8]. Although verteporfin can aggregate in glioblastoma cells [32] and has a photodynamic effect leading to extensive protein cross-linking, this would not be applicable in non-illuminated solid tumors. 

## 5. Conclusions

This study provides further evidence that verteporfin-induced autophagy, independent of its photodynamic property, plays a role in antagonizing CAR-induced trogocytosis and might also be promoting the phagocytic activity of CAR T cells. By inducing autophagy to allow clearance of trogosome cargo, verteporfin counteracts the recycling of cargo to the cell membrane. Given its trogocytosis-dependent and -independent drug actions, verteporfin may be particularly effective in counteracting both tumor antigen and PD-L1 trogocytosis in PD-L1+ target tumor cells [26].

## Figures and Tables

**Figure 1 cancers-15-01085-f001:**
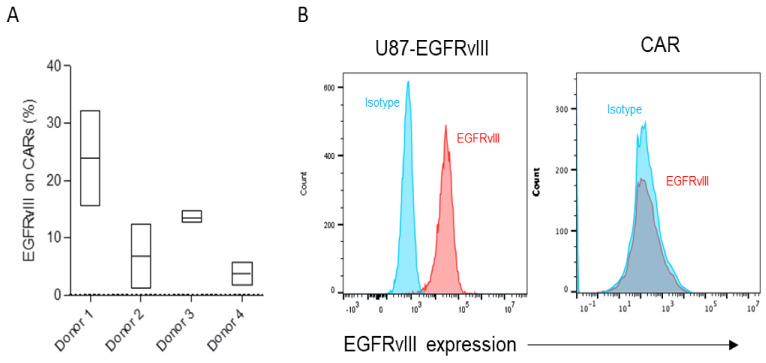
(**A**,**B**) Trogocytic EGFRvIII expression was gated on CAR+ cells analyzed in triplicate following co-culture with targets (U87-EGFRvIII) for 2 h. The gate was set based on CAR T cells without a target and then plotted based on the range of expression on flow cytometry. (**B**) A representative flow cytometry analysis of baseline EGFRvIII expression on target tumor (U87-EGFRvIII) and EGFRvIII CAR T cells.

**Figure 2 cancers-15-01085-f002:**
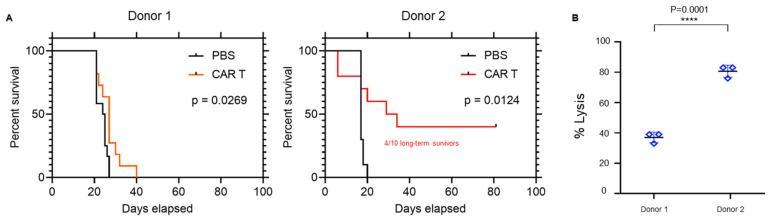
Donor differences in CAR activity and trogocytosis. (**A**) Representative survival curves of mice bearing U87-EGFRvIII tumors treated with EGFRvIII CARs engineered from Donor 1 and Donor 2 (n = 10/group). (**B**) Target tumor cell (U87-EGFRvIII) lysis following co-culture with EGFRvIII CARs (E:T 5:1) from Donors 1 and 2 for 24 h. **** *p* = 0.0001.

**Figure 3 cancers-15-01085-f003:**
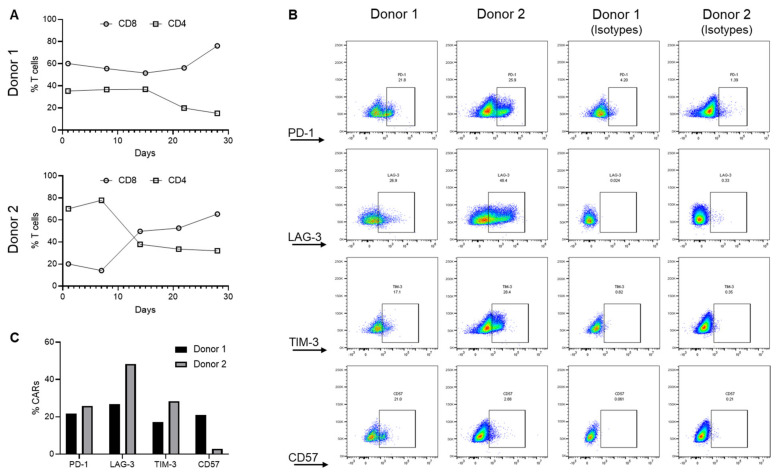
Donor lineage skewing and the expression of exhaustion markers. (**A**) Changes in CD8 and CD4 cells during ex vivo expansion of EGFRvIII CAR T cells. (**B**,**C**) Expression of T cell exhaustion (PD-1, TIM3, and LAG3) and senescence (CD57) markers on CAR T cells. These data summarize the analysis generated from two different donors (Donor 1 and Donor 2) and are representative of a triplicate analysis.

**Figure 4 cancers-15-01085-f004:**
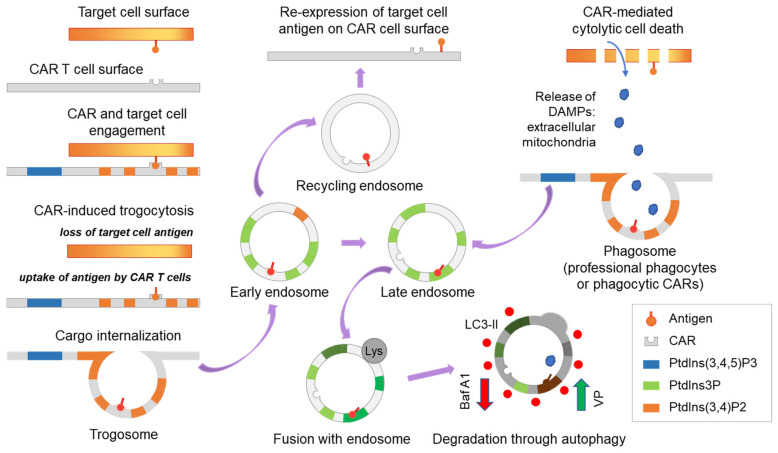
Scheme demonstrating CAR-induced trogocytosis. Trogocytosis occurs upon CAR-target engagement resulting in the expression of the target tumor antigen on CAR T cells. Following extraction by CAR T cells, the tumor antigen is internalized and processed through the endosome system and then is either recycled to the cell surface or shuttled to late endosomes and subsequently degraded through lysosome and autophagy-dependent mechanisms. Activation and inhibition of autophagy may regulate trogocytosis and modulate CAR T cell function. In addition, trogocytosis may interface with phagocytosis. Baf A1, bafilomycin A1, an inhibitor of vacuolar ATPase and autophagy; VP, verteporfin, an activator of selective autophagy; DAMPs, damage-associated molecular patterns.

**Figure 5 cancers-15-01085-f005:**
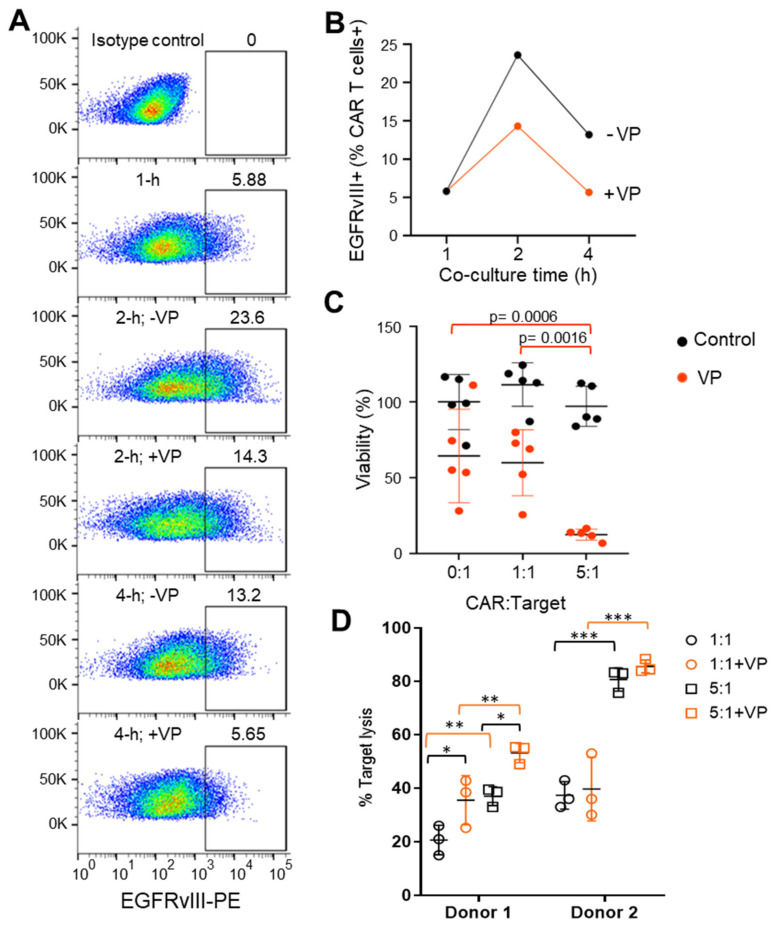
Verteporfin inhibits CAR-induced trogocytosis and increases CAR activity. (**A**) Representative flow cytometry example of EGFRvIII expression on Donor 1 CAR prior to or following co-culture with targets (U87-EGFRvIII) in the absence (-) or presence (+) of verteporfin (VP). The gate is set against the isotype control and was conducted in triplicate. (**B**) Summarized kinetic data from A of CAR T cell EGFRvIII expression in the absence and presence of VP. (**C**) Target (U87-EGFRvIII) cell viability treated with Donor 1 CARs in the absence (control; black circles) or presence of verteporfin (VP; red circles) for 24 h. A Two-way ANOVA test and Turkey’s multiple comparisons were conducted with no statistical significance between the control groups (black circles). Statistical significance between the VP-treated groups (red circles): *p* = 0.0006 (0:1 vs. 5:1), *p* = 0.0016 (1:1 vs. 5:1). (**D**) CAR-mediated lysis of target (U87-EGFRvIII) cells treated with EGFRvIII CARs derived from Donor 1 and Donor 2 at 1:1 and 5:1 ratio in the absence (-VP) or presence (+VP) of verteporfin for 24 h. * *p* < 0.05; ** *p* < 0.01; *** *p* < 0.001 (unpaired *t*-test). For Donor 1: *p* = 0.039 (1:1 vs. 1:1+VP), *p* = 0.0312 (5:1 vs. 5:1+VP), *p* = 0.0145 (1:1 vs. 5:1), *p* = 0.0117 (1:1+VP vs. 5:1+VP). For Donor 2: not significant (1:1 vs. 1:1+VP), not significant (5:1 vs. 5:1+VP), p = 0.0001 (1:1 vs. 5:1), *p* = 0.0001 (1:1+VP vs. 5:1+VP).

**Figure 6 cancers-15-01085-f006:**
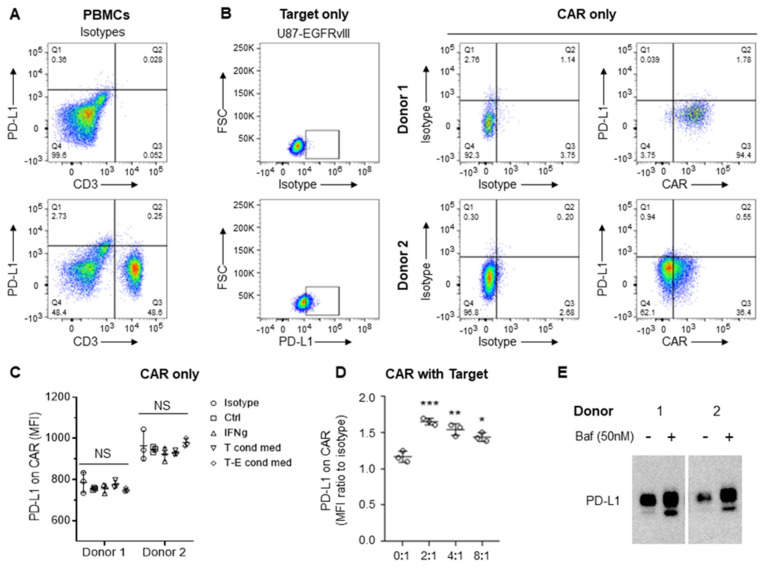
Expression of PD-L1 on target tumor but not on CAR T cells. (**A**) Flow cytometry analysis of baseline PD-L1 expression on CD3+ primary T cells from peripheral blood mononuclear cells (PBMCs). Analysis was conducted in triplicates. (**B**) Flow cytometry analysis of target tumor (U87-EGFRvIII) and EGFRvIII CAR T cells. (**C**,**D**) Summary of flow cytometry analysis of PD-L1 levels (MFI, mean fluorescence intensity) on CAR T cells expanded on anti-CD3/CD28 dynabeads (Ctl), treated with interferon γ (IFNγ, 5 ng/ml), exposed to condition media from the target (T cond med) or target-CAR (T-E = 2:1) conditioned media (**D**). Summarized flow cytometry analysis of PD-L1 on CARs after co-culture with target tumor for 20 h, taken from biological replicates of Donor 1. Targets and CARs are distinct populations gated on SCC/FSC, GFP (targets), and CAR (CD3). * *p* < 0.05; ** *p* < 0.01; *** *p* < 0.001 relative to no target. (**E**) Western blot analysis of PD-L1 expression on EGFRvIII-specific CARs generated by Donors 1 and 2 following CAR-target co-culture for 20 h in the absence (-) or presence (+) of bafilomycin (Baf). Original flow cytometry see Appendix A. Original blots see Appendix A.

**Figure 7 cancers-15-01085-f007:**
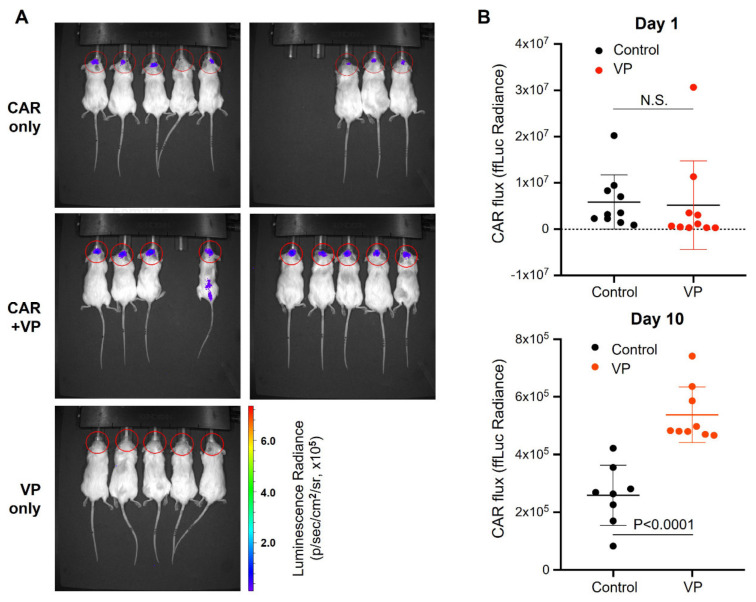
Verteporfin increases CAR persistence in tumors. (**A**) BLI of the CAR signals was assayed on day 10 after infusion with and without verteporfin. (**B**) Summarized data of firefly bioluminescence (ffLuc) imaging of CAR signal acquired on day 1 (infusion) and day 10 after infusion into U87-EGFRvII tumors (implanted on day 4) with and without treatment of verteporfin for 3 days prior to CAR infusion.

**Figure 8 cancers-15-01085-f008:**
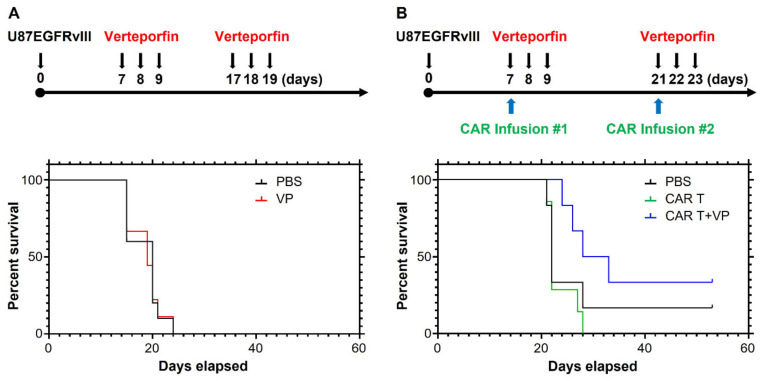
Verteporfin reverses CAR dysfunction. (**A**) Treatment schema and survival curves of mice bearing U87-EGFRvII tumor treated with or without verteporfin, median survival 17 days (PBS and VP). (**B**) Treatment schema and survival curves of mice bearing U87-EGFRvIII tumors treated with Donor 1 CAR with or without verteporfin, median survival 22 (PBS), 22 (CAR T), and 30.5 days (CAR T+VP). If the mouse was moribund or dead, it did not receive the second CAR treatment.

## Data Availability

All data is presented in this article.

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
