# Peer review of "A Case Study of Chimeric Antigen Receptor T Cell Function: Donor Therapeutic Differences in Activity and Modulation with Verteporfin"

_cancers, 2023, doi:10.3390/cancers15041085_

Round 1

Reviewer 1 Report

Authors undertaken investigations of  Chimeric Antigen Receptor (CAR) T Cell function in the therapy of intracranial glioma. The exploration of CAR T-cell therapy in GBM has begun and many investigations have been published. Particular problem in that treatment is due to CAR T cell dysfunction  caused by the extraction and expression of tumor antigens through trogocytosis. It has been recognized that the antygen EGFRvIII, represent the CAR target for the human CAR T-cell studies reported in GBM.  Investigated were models in vitro human engineered epidermal growth factor receptor variant III (EG- 21 FRvIII) CAR T cell line and specific CAR T cells targeting glioma generated from various donors and in vivo murine glioma model.  Autophagy anti- cancer activity of that cells was evaluated in the presence of inducer (verteporfin) or inhibitor (bafilomycin A1) of that process.

In vivo and in vitro experiments were well planned and performed and applied methods, properly described, allowed to obtain interesting results, well documented and visualized. Presented data are important for both:  possibile clinical application of pharmacologicaly induced  CAR T cells autophagy in glioma therapy  as well as for  accumulation of the new data concerning interaction of that two counteracting processes; autophagy and trogocytosis in CAR T cells under various pharmacological stimulations.  Interesting is information that verteporfin-induced autophagy beside a role in antagonizing CAR-induced trogocytosis  might also promoting phagocytic activity of CAR T cells.  Particularly important is observation of the cross-cell transfer of the immune checkpoint ligand PD-L1 from tumor cells to CAR T cells,  possibly the first description of PD-L1 expression on CAR T cells acquired through trogocytosis. Autors made note that verteporfin may have additional multiple biological and cellular  effects including alteration of the CAR turnover or proliferation so its clinical application requires further studies.

Manuscript should be published after including certain import ant citations in the Introduction and Discusson :

Boccalatte F et al. Cancers (Basel). 2022,14;20;5108,

Bagley SJ . et al.  J Transl Med. 2020,18;428,

Karschnia P. et al.  Neurology.2021,97;5;218-230,

Di Cintio F et al. Front. Neurosci. 2020,14:603647,

Calori I.R.et al. Dyes and Pigments, 2020,182;108598,

Maggs L et al. Front. Neurosci. 2021,15:662064,

Prapa, M. et al. Precis. Onc. 2021,5; 93.

Author Response

We thank the reviewer for his acknowledgments and comments. All of the aforementioned articles have been included in the Introduction or Discussion.

Reviewer 2 Report

This study proposes CAR-induced autophagy as a mechanism to counteract CAR-induced trogocytosis.

Revisions:

- Informed consent and ethical committe information should be added. 

- Was normality of the samples studied before applying any statistical test? It should be added in the method section.

- The main limitation of this study is that only 2 donors were studied. More donors would be convenient to strength this work.

- Do flow cytometry experiments have repetitions? They should be added in the figures.

Author Response

We thank the reviewer for his comments. Please find our revisions below.

- Informed consent and ethical committee information should be added.

Revision: The blood was purchased from a blood bank to create the CAR T cells and as such informed consent is not necessary/possible. A statement regarding approval for the animal studies has been included.

- Was normality of the samples studied before applying any statistical test? It should be added in the method section.

Revision: Data was examined for normality through boxplots and QQ plots prior to application of linear statistical models. This information was added accordingly to the methods section.

- The main limitation of this study is that only 2 donors were studied. More donors would be convenient to strength this work.

Revision: The screening for trogocytosis activity was done on four donors which has now been included in Fig. 1A. 

- Do flow cytometry experiments have repetitions? They should be added in the figures.

Revision: The flow cytometry experiments were done in triplicates and each figure legend contains the number of replicates.

Reviewer 3 Report

In my opinion this preclinical study is very interesting, exploring the significance of a newly discovered process such as trogocytosis. Inhibiting this phenomenon may be an attractive strategy to counteract CAR-T resistance.

Just few comment and clarifications:

Introduction (line 40)

Briefly introduce what CAR-T are

Lines 52-55 // lines 64-68

I think it is better not to add the final results of the study in the introduction but to specify and explain it in the discussion

Lines 58-61

Please, explain better and more clearly the period about CAR-T killing and autophagy meaning in this contest

Line 138

Please, define the meaning of “where indicated”

Lines 153-157

 Please, define better in  methods

Lines 174-176

Please clarify the period

Lines 249-259

I'm not sure it's of interest in this article

Figure 4

Please, simplify the caption

Author Response

We thank the reviewer for his acknowledgments and inputs that ameliorated the quality of our paper.

Introduction (line 40): Briefly introduce what CAR-T are

Revision: A description has been provided along with an associated reference.

Lines 52-55 // lines 64-68: I think it is better not to add the final results of the study in the introduction but to specify and explain it in the discussion

Revision: This content has been relocated to the Discussion section as suggested.

Lines 58-61: Please, explain better and more clearly the period about CAR-T killing and autophagy meaning in this context

Revision: We agree with the reviewer that these two sentences are confusing and have clarified this paragraph.

Line 138: Please, define the meaning of “where indicated”

Revision: This has been removed as it is evident in each figure.

Lines 153-157: Please, define better in methods

Revision: This section has been redone to clarify the selection process of the CARs was based on differences in trogocytosis.

Lines 174-176: Please clarify the period

Revision: Trogocytosis was measured at 2 hours and this information has been added to the legend.

Lines 249-259: I'm not sure it's of interest in this article

Revision: This point has been truncated.

Figure 4: Please, simplify the caption

Revision: We have complied with this request.

Reviewer 4 Report

Liang et al. in their preliminary report describe potential role of CAR T cells' trogocytosis in T cell therapy of gliomas. The manuscript is well written and data presented with clarity. My main criticism is that the presented work relates to just two tumor examples hence the data are truly preliminary. This also raises the question about the generality of the observed phenomena, 

1) The U-87 cells may not be the best model for such studies as their origin has been questioned (Sci Transl Med 8(354):354re3) and the over-expression of the receptor is not wide-spread (Fig. 1A).

2) Donors 1 and 2 CD8 count was very different (Fig. 3).

3) The schemata shown in Fig. 4 should be better placed in either Methods or Discussion section.

Author Response

We thank the reviewer for his comments and acknowledgments of our work.

  • The U-87 cells may not be the best model for such studies as their origin has been questioned (Sci Transl Med 8(354):354re3) and the over-expression of the receptor is not wide-spread (Fig. 1A).

Response: We acknowledge the limitation of the U87 model. However, this cell line is extensively used in almost 2000 PubMed publications including in Cancers (Kuo YH, et al. Cancers. 2022; Sun W, et al. Cancers. 2022; Ruggiero MR, et al. Cancers. 2022; Sun Z, et al. Cancers. 2022; Zev A. Binder et al. Cancer Cell. 2018; Mathews J et al, Cancers. 2022; Filippone A et al.  Cancers. 2022; Chang HH et al. Cancers. 2022; Biscop E et al, Cancers. 2019; etc.). Given that CAR T cells rely on cytotoxic mechanisms distinct from the effect induced by radiation and chemotherapy, the U87-EGFRvIII line is a valuable resource for studies, with almost 30 PubMed publications, involving targeting with EGFRvIII-specific CAR T cells. Patient-derived xenograft overexpressing EGFRvIII will be considered in our future extended studies.

  • Donors 1 and 2 CD8 count was very different (Fig. 3).

Response: The blood was purchased from the Gulf Coast Regional Blood Bank and variability in the amounts of immune populations between otherwise healthy donors exist (Burnham et al. 2020; O’Reilly et al. 2022) and unfortunately cannot be predicted. Also, variability in the purification and permeability of the Ficoll-paque isolation of immune cells plays a role (Puleo et al. 2017).

  • The schemata shown in Fig. 4 should be better placed in either Methods or Discussion section.

Revision: In the discussion, we mentioned: “…Our findings suggest that trogocytosis can be counteracted by verteporfin-facilitated autophagy, pushing the downstream pathway into late-endosome formation and degradation of the target cell antigens instead of its re-expression on the cell surface by recycling endosomes (Fig. 4), …”

Round 2

Reviewer 4 Report

The authors clarified the raised issues.